# Iron deposition in autopsied liver specimens from older patients receiving intravenous iron infusion

**Hiroyasu Akatsu**[1,2,3]*, **Toshie Manabe**[4], **Yoshihiro Kawade**[1], **Hajime Tanaka**[1], **Takayoshi Kanematsu**[1], **Kazuyuki Arakawa**[1], **Yoshiyuki Masaki**[1], **Chie Hishida**[2], **Takeshi Kanesaka**[2], **Norihiro Ogawa**[2], **Yoshio Hashizume**[2], **Koichi Tsuneyama**[5], **Hirotaka Ohara**[1], **Mitsuo Maruyama**[3], **Takayuki Yamamoto**[2]

**1** Department of Community-based Medical Education, Nagoya City University Graduate School of Medical Sciences, Nagoya, Japan, **2** Fukushimura Hospital, Toyohashi, Japan, **3** Department of Mechanism of Aging, Research Institute, National Center for Geriatrics and Gerontology, Obu, Japan, **4** Division of Community and Family Medicine, Center of Community Medicine, Jichi Medical University, Shimotsuke, Japan, **5** Department of Pathology, Tokushima University School of Medicine, Tokushima, Japan

* akatu@med.nagoya-cu.ac.jp, akatu@chojuken.net

## Abstract

### Background

Vitamins and minerals are routinely administered by total parenteral nutrition (TPN). However, in Japan, adjustments in iron dosage are difficult because blended mineral preparations are often used. It is therefore unclear whether the iron content is appropriate in cases of long-term TPN. The aim of the study was to assess the influence of iron administration by long-term TPN on iron deposition in post-mortem liver samples isolated from older deceased patients.

### Methods

Liver tissues were collected from post-mortem autopsies of 187 patients over a period of 15 years. Samples were stained with Prussian blue and histologically evaluated from Grade 0–V by at least three different observers. Specimens with positive and negative iron staining were compared, and positive samples were grouped according to the level and distribution of the staining. Post-mortem blood obtained from the subclavian vein during autopsy was also analysed. Samples were collected for the measurement of unsaturated serum iron, serum iron, albumin, prealbumin, hepcidin, and IL-6 concentrations.

### Results

Iron accumulation in the liver was significantly higher in male patients (p = 0.005) with a history of surgery (p = 0.044) or central vein administration of iron (p<0.001). Additionally, the duration of TPN in the iron-positive group was significantly longer than in the iron-negative group (p = 0.038). Serum analysis revealed that unsaturated serum iron was significantly higher in the iron-negative group and that ferritin and serum iron were significantly higher in the iron-positive group. No other statistically significant differences were observed between the two groups.

**Data Availability Statement:** All relevant data are included within the paper and its Supporting Information file.

**Funding:** Our study was supported by the research grant from Longevity Sciences (ID 29–26) from the National Center for Geriatrics and Gerontology (NCGG), Obu, Japan to HA. The funder had no role in study design, data collection and analysis, decision to publish, or preparation of the manuscript.

**Competing interests:** The authors have declared that no competing interests exist.

## Conclusions

Chronic intravenous administration of iron was associated with iron deposition in the liver, even when given the minimum recommended dosage. In long-term TPN patients, the iron dose should therefore be carefully considered.

## Introduction

During long-term total parenteral nutrition (TPN), high doses of certain water-soluble vitamins, such as those in the vitamin B family, do not pose a significant health risk. However, minerals are absorbed into the body unregulated and are not readily excreted during TPN. The accumulation of minerals, such as iron, is difficult to assess during TPN administration. One solution is to monitor blood mineral levels after administering a single dose; however, blood concentrations may not be an accurate reflection of iron accumulation in tissues.

Iron is the most abundant trace mineral in the body and plays a critical role in oxygen transport. When iron is low, a number of health issues arise, such as anaemia. Problems also occur when iron levels are high, and the Fenton reaction, which requires iron, may cause oxidative stress and lead to excessive lipid oxidation, DNA damage, and cell death by apoptosis, and carcinogenesis [1].

During intravenous iron administration, such as TPN, unlimited amounts of iron can be absorbed. This excess iron accumulates and leads to tissue damage. There are recommended safe doses of iron in Europe and the United States [2]. In patients with chronic inflammatory gastrointestinal disease, congenital problems in the digestive tract, or intestinal deficiencies secondary to surgery, iron is often administered intravenously.

Though uncommon in other counties, artificial nutrition, especially intravenous nutrition, is often the preferred treatment for ageing patients in Japan, particularly those who have problems with oral intake of food caused by dysphagia or dementia. Iron is often administered during TPN. Blended mineral preparations have been developed for this purpose, and it is therefore common for iron to be administered intravenously without strict monitoring of mineral levels.

Until the last century, long-term TPN was limited to specific patients, such as adults with short bowel syndrome. In Japan, TPN with a premixed blend of trace elements is still the treatment of choice for the management of older terminal patients. A study in Japan reported that, over the course of one year, half of the patients who died or were discharged received artificial nutrition and more than half received TPN on average for a period of 200 days [3]. However, the association between iron administration by TPN and iron deposition in the liver has never been examined.

The aim of the present study was to assess iron deposition in post-mortem liver samples isolated from older deceased patients to evaluate the influence of iron administration by long-term TPN.

## Materials and methods

### Study design and patient population

We conducted a retrospective observational study using data on blood indices, clinical history, and diagnosis prior to death for 187 patients, who were hospitalized and autopsied from 1999 to 2014 in Fukushimura Hospital, Aichi, Japan.

## Clinical records and laboratory data

From the clinical records, detailed data on intravenous iron administration, including volume, date, periods, and frequency, were assessed. Data on the number of days and the volume of iron administration were collected for one year prior to the patients' death and during the entire period of hospitalization. Laboratory data on iron metabolism, inflammatory factors, liver function, kidney function, and blood cell count were also collected. Several patients had hepatitis B or C virus antibodies, but did not have chronic hepatitis.

## Blood sampling at autopsy and biochemical analysis

Dissections were performed immediately after death (if the patient died after midnight, the corpse was stored at 4°C until dissection), and blood was sampled from the subclavian vein before skin incision. Blood was centrifuged within 3 hours of collection and the serum was stored at −80°C until analysis. From these samples, aspartate aminotransferase (AST), alanine aminotransferase (ALT), unsaturated iron blood concentration (UIBC), serum iron, and ferritin were analyzed by Tosan Labo Center (Toyohashi, Japan), and albumin, pre-albumin, hepcidin, and interleukin-6 (IL-6) were analyzed by Hoken-Kagaku, Ltd. (Osaka, Japan). As shown in Tables 4 and 5, the number of samples varied from patient-to-patient because the volume of blood was insufficient to analyze all parameters.

## Liver tissue sampling and staining for iron

Whole abdominal organs were removed according to standard dissection procedures. The liver was then separated from the abdominal tissue and immersed in buffered formalin for 1–2 weeks. If no lesions were observed, samples of the liver were obtained from segments 3 and 8, 15 mm$^3$ in size. Then, according to standard protocols, the tissue was embedded in paraffin wax. The paraffin-embedded liver was sectioned into 4.5-micron slices, followed by deparaffinisation and rehydration. Prussian blue was used to stain iron histologically, while haematoxylin was used to stain nuclei.

## The grading of iron deposition

Three investigators (H.A., K.T., H.O.), who specialize in the study of liver or liver pathology, evaluated the Prussian blue staining in each liver sample independently and recorded their findings on a custom data collection form. We verified the accuracy of these evaluations by comparing the forms of each investigator and resolving any discrepancies through discussion. The staining intensity was graded from a scale of 0 to V: Grade 0 (minimum grade), no staining; Grade I, small number of positive Kupffer cells; Grade II, moderate number of positive Kupffer cells; Grade III, most Kupffer cells are positive, with some positive hepatocytes, or no staining of Kupffer cells, but some positive hepatocytes; and Grade IV, large number of positive Kupffer cells and hepatocytes; Grade V, most hepatocytes and Kupffer cells are positive.

## Statistical analyses

Data are reported as percentages for categorical variables, and as medians with interquartile range (IQR: 25%–75%) or as means with standard deviation (SD) for continuous variables. Comparisons were made between specimens with positive- and negative-iron staining and among grades of iron staining for each variable, using the $\chi^2$ test or Fisher's exact test for categorical variables, and Mann–Whitney $U$ test, Kruskal–Wallis test, one-way analysis of variance, paired $t$-test, or Wilcoxon signed-rank test for continuous variables. Additional factors related to iron staining were determined using logistic regression analysis with independent

variables, including gender and age, if p was < 0.05 by univariate analysis. A step-wise selection method was used to select variables.

Data were analyzed using IBM SPSS Statistics version 25.0 (IBM, Armonk, NY, USA). For all analyses, statistical tests were two-tailed, and $p < 0.05$ was considered statistically significant.

### Ethical statement

The study was approved by the Ethics Committee of Fukushimura Hospital (approval number 353; Sept. 29, 2015). Informed consent from patients and their families was obtained for all human samples collected, with the knowledge that tissues taken during autopsy would be used for experimentation and analysis.

## Results

### Clinical characteristics and cause of death

The clinical characteristics of all patients according to positive or negative iron staining are shown in Table 1.

Among the iron-positive patients, the vast majority were male (p = 0.005) and were more likely to have a history of surgery (p = 0.044). The most frequent cases of surgery were for femoral fracture (10 cases), female reproductive ailments (7 cases) or digestive system disorders outside of the intestine (7 cases). There were two cases of bowel resection for colorectal cancer, and both patients had an iron deposition grade of I or II in the liver. Patients with a history of fractures, other than a femoral fracture, were more likely to be iron-negative. There were no other factors significantly associated with iron deposition in the clinical history or diagnosis.

In Table 2, the cause of death at autopsy is compared with our findings on iron deposition. No correlation was found between cause of death and liver iron deposition.

**Table 1. Comparison of general characteristics of patients with and without iron staining.**

|  | Without iron staining (n = 92) | With iron staining (n = 95) | *p* value |
|---|---|---|---|
| **Sex**, Male, n (%) | 33 (35.9) | 54 (56.8) | 0.005 |
| **Age at death**, Median (IQR), y | 86 (78–93) | 85 (77–95) | 0.513 |
| **Clinical history** |  |  |  |
| Hospitalized transfusion | 5 (5.4) | 6 (6.3) | 1.000 |
| Transfusion | 3 (3.3) | 2 (2.1) | 0.679 |
| Operation | 23 (25.0) | 37 (38.9) | 0.044 |
| Trauma | 13 (14.1) | 10 (10.5) | 0.509 |
| Fracture | 23 (25.0) | 11 (11.6) | 0.022 |
| **Underlying disease or condition** |  |  |  |
| HBV ab+ | 2 (2.2) | 3 (3.2) | 1.000 |
| HCV ab+ | 4 (4.3) | 7 (7.4) | 0.537 |
| Cerebral infarction | 35 (38.0) | 24 (25.3) | 0.083 |
| Malignant neoplasm | 24 (26.1) | 21 (22.1) | 0.608 |
| Liver cirrhosis | 2 (2.2) | 3 (3.2) | 1.000 |
| Diabetes mellitus | 15 (16) | 9 (9.5) | 0.193 |
| Hypertension | 25 (27.2) | 16 (16.8) | 0.112 |
| Dementia | 41 (44.6) | 47 (49.5) | 0.559 |

IQR, interquartile range; HBV ab+, hepatitis B virus antibody positive; HCV ab+, hepatitis C virus antibody positive.

**Table 2. Comparison of cause of death in patients with and without iron staining.**

|  | Without iron staining (n = 92) | With iron staining (n = 95) | p-value |
|---|---|---|---|
| Heart failure | 25 (27.2) | 15 (15.8) | 0.074 |
| Cardiovascular disease | 2 (2.2) | 4 (4.2) | 0.683 |
| Digestive organ disorder | 3 (3.3) | 1 (1.1) | 0.363 |
| Malignant neoplasm | 11 (12.0) | 8 (8.4) | 0.425 |
| Pneumonia | 23 (25.0) | 29 (30.5) | 0.419 |
| Respiratory failure | 11 (12.0) | 18 (18.9) | 0.227 |
| Renal failure | 3 (3.3) | 5 (5.3) | 0.721 |
| Liver failure | 2 (2.2) | 3 (3.2) | 1.000 |
| Sepsis | 5 (5.4) | 2 (2.1) | 0.273 |
| Multi-organ failure | 5 (5.4) | 10 (10.5) | 0.282 |
| Sudden death | 5 (5.4) | 3 (3.2) | 0.492 |

## Iron deposition in liver

Of the 187 cases evaluated, more than half (95) had positive Prussian blue staining, indicative of iron accumulation. Moreover, in 128 of the cases (68%), patients had been receiving TPN at the time of death. The staining patterns are shown in Fig 1.

The distribution of the number of cases with iron staining was 92 in Grade I (Fig 1B), 27 in Grade II (Fig 1C), 14 in Grade III (Fig 1D or 1E), and 6 in Grade IV (Fig 1F), and 4 in Grade V (Fig 1F). Only one sample, from a patient who had been diagnosed with haemochromatosis (details unknown), had an extremely high level of staining (iron-positive hepatocytes and almost all Kupffer cells were positive) and was classified as Grade V (Fig 1G).

## Nutritional administration

Table 3 compares the modes of nutritional administration and the status of TPN with positive or negative iron staining in patients.

The percentage of patients receiving TPN versus all other modes of artificial nutrition was significantly higher in the iron-positive group (82.1%; $p < 0.001$) compared with the iron-negative group. The number of patients with iron-positive staining who had been receiving TPN at the time of death was 128, although as many as 140 patients had received artificial nutrition in the year before death.

The number of days of TPN administration was significantly higher in the iron-positive group compared with the iron-negative group both in the year before death ($p = 0.004$) and during hospitalization ($p = 0.038$). However, the volume of TPN was not significantly different between the groups in the year before death ($p = 0.222$) or during the hospitalization ($p = 0.220$).

## Biochemical analysis of blood

Blood chemistry (iron-related factors, hepatobiliary enzymes), hepcidin, IL-6, albumin, and prealbumin were measured from blood collected at the time of autopsy. UIBC was significantly higher in the iron-negative group, and ferritin and iron were significantly higher in the iron-positive group (Table 4).

The blood data from three patients in the iron-positive group was compared and is shown in Table 5.

UIBC was significantly lower in cases with stronger iron deposition ($p < 0.001$), and ferritin ($p < 0.001$) and iron ($p = 0.001$) were significantly higher in cases with stronger iron deposition.

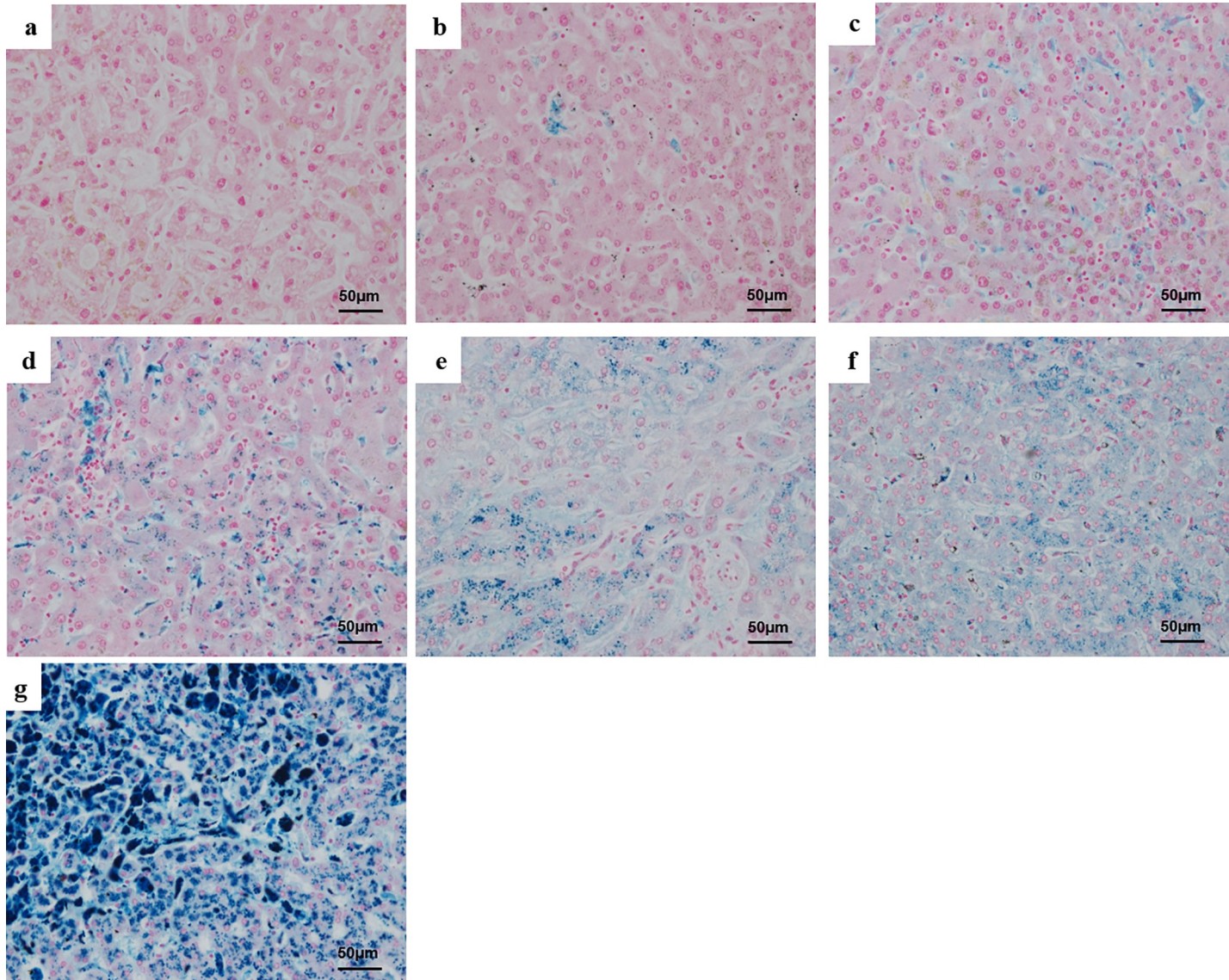

**Fig 1. Prussian blue staining in autopsied liver.** a, Grade 0 (no staining). b, Grade I (small number of positive Kupffer cells). c, Grade II (moderate number of positive Kupffer cells). d, Grade III (most Kupffer cells are positive, with some positive hepatocytes). e, Grade III (no staining of Kupffer cells, but some positive hepatocytes). f, Grade IV (large number of positive Kupffer cells and hepatocytes). g, Grade V (most hepatocytes and Kupffer cells are positive).

### Factors associated with iron staining

Logistic regression analysis revealed that male sex (odds ratio [OR], 4.119; 95% confidence interval [CI], 1.773–9.566) and iron in post-mortem serum (OR, 1.025; 95% CI, 1.010–1.041) were factors associated with iron staining if the volume of TPN was included in the adjusted variables (Table 6). If the volume of TPN was excluded from this analysis, UIBC in post-mortem serum (OR, 1.014; 95% CI, 1.003–1.022) and administration of TPN (OR, 3.586; 95% CI, 1.664–7.732) were factors significantly associated with iron staining (Table 6).

## Discussion

The present study revealed that iron administration by long-term TPN significantly affected iron deposition in the liver of older patients, even at the minimum recommended iron dosage.

**Table 3. Conditions of artificial nutritional support and iron administration in post-mortem liver with or without iron staining.**

| | Without iron staining (n = 92) | With iron staining (n = 95) | p value |
|---|---|---|---|
| **Route of artificial nutrition at death** | | | <0.001 |
| NG | 2 (2.2) | 0 (0.0) | |
| PEG | 5 (5.4) | 2 (2.1) | |
| Oral intake | 3 (3.3) | 0 (00) | |
| TPN | 50 (54.3) | 78 (82.1) | |
| PPN | 19 (20.7) | 14 (14.7) | |
| Unknown | 13 (14.1) | 1 (1.1) | |
| **Conditions of iron administration**, median (IQR) | | | |
| Number of days of TPN in the year before death, days (n = 140) | 41 (14–104) | 97 (31–243) | 0.004 |
| Volume of iron administered in the year before death, μmol (n = 132) | 805 (166–2231) | 13512 (245–3421) | 0.222 |
| Number of days of iron infusion during hospitalization, days (n = 161) | 46 (22–130) | 99 (32–236) | 0.038 |
| Total volume of iron infused during lifetime (μmol) (n = 157) | 1155 (315–2887) | 1907 (315–4541) | 0.220 |

NG, naso-gastric tube; PEG, percutaneous endoscopic gastrostomy; TPN, total parenteral nutrition; PPN, partial parenteral nutrition.

This is the first study to demonstrate an association between iron administration and iron deposition in post-mortem liver specimens.

The average adult loses approximately one mg of iron per day, mainly because of cell turn-over [4]. About two-thirds of iron is lost in the gastrointestinal tract, and one-third is lost from the skin. Iron loss in urine and sweat is minimal because there is no physiological mechanism for iron excretion.

With oral intake of iron, about two-thirds is absorbed as haeme via haeme carrier protein 1 (HCP1) [5]. The remaining one-third of iron is absorbed as non-haeme $Fe^{3+}$, which is reduced by duodenal cytochrome b (Dcytb) [6, 7] to $Fe^{2+}$, and then is absorbed by divalent metal transporter 1 (DMT1) [7, 8]. This absorption mechanism is thought to be restricted and adjusted based on the body's existing iron levels. However, because iron absorption also occurs in the

**Table 4. Laboratory indices in patients with and without iron staining using post-mortem serum.**

| Median (IQR) | Without iron staining (n = 92) | With iron staining (n = 95) | p value |
|---|---|---|---|
| AST (IU/L) (n = 169) | 84.0 (41.0–291.0) | 70.5 (36.0–190.0) | 0.397 |
| ALT (IU/L) (n = 169) | 28.5 (7.8–103.8) | 29.0 (9.0–48.5) | 0.382 |
| γ-GTP (IU/L) (n = 169) | 40.0 (20.0–61.0) | 34.0 (20.0–75.8) | 0.963 |
| LDH (IU/L) (n = 169) | 270.0 (206.0–537.3) | 321.0 (150.5–394.0) | 0.150 |
| ALP (IU/L) (n = 169) | 335.0 (251.0–596.0) | 379.5 (250.5–521.0) | 0.756 |
| UIBC(μg/dL) (n = 171) | 94.0 (50.0–121.0) | 40.5 (24.0–70.3) | <0.001 |
| Ferritin (ng/mL) (n = 171) | 637.0 (266.0–1429.0) | 1281.5 (708.8–2518.5) | <0.001 |
| Fe (μg/dL) (n = 171) | 49.0 (39.0–75.0) | 64.0 (45.3–94.8) | 0.002 |
| Hepcidin (n = 49) | 50.7 (34.2–151.9) | 118.9 (50.1–176.6) | 0.169 |
| IL-6 (n = 49) | 7.3 (5.7–9.0) | 7.2 (6.2–9.1) | 0.909 |
| Pre-alb (n = 169) | 5.9 (3.3–10.0) | 4.2 (2.6–7.8) | 0.844 |
| alb (n = 171) | 1.8 (1.4–2.2) | 1.5 (1.2–1.9) | 0.440 |

IQR, interquartile range; AST, aspartate transaminase; ALT, alanine transaminase; γ-GTP, γ-glutamyl transpeptidase; ALP, alkaline phosphatase; UIBC, unsaturated iron binding capacity; Fe, serum iron; IL-6, interleukin 6; pre-alb, pre-albumin; alb, albumin.

**Table 5. Laboratory findings in post-mortem serum according to iron staining grade.**

| Median (IQR) | Grade of iron staining | | | |
|---|---|---|---|---|
| | Grade 0 (n = 92) | Grade I (n = 44) | ≥ Grade II (n = 51) | *p* value |
| **Sex, Male, n (%)** | **33 (35.9)** | **20 (45.5)** | **34 (66.7)** | **0.002** |
| AST (IU/L) (n = 169) | 84.0 (41.0–291.0) | 93.5 (40.0–292.3) | 53.0 (31.5–141.0) | 0.106 |
| ALT (IU/L) (n = 169) | 36.0 (11.0–104.0) | 30.0 (12.0–69.3) | 23.0 (12.0–69.3) | 0.575 |
| γ-GTP (IU/L) (n = 169) | 40.0 (20.0–62.0) | 37.5 (19.0–75.8) | 30.0 (12.0–69.3) | 0.995 |
| LDH (IU/L) (n = 169) | 380.0 (220.0–767.0) | 392.5 (253.0–548.8) | 281.5 (166.0–460.3) | 0.043 |
| ALP (IU/L) (n = 169) | 335.0 (251.0–460.3) | 380.0 (259.8–515.0) | 374.5 (248.3–656.5) | 0.952 |
| UIBC (μg/dL) (n = 171) | 94.0 (50.0–121.0) | 51.0 (27.0–74.5) | 25.0 (24.0–65.0) | <0.001 |
| Ferritin (ng/mL) (n = 171) | 637.0 (266.0–1429.0) | 1134.0 (550.5–2477.0) | 1434.0 (780.0–2524.0) | <0.001 |
| Fe (μg/dL) (n = 171) | 49.0 (39.0–75.0) | 59.0 (41.5–75.0) | 73.0 (57.0–103.0) | 0.001 |
| Hepcidin (g/dL) (n = 49) | 50.7 (34.2–151.9) | 125.3 (50.1–157.7) | 111.7 (48.1–188.1) | 0.377 |
| IL-6 (pg/mL) (n = 49) | 7.3 (5.7–9.0) | 7.7 (6.6–8.5) | 7.0 (5.9–9.4) | 0.477 |
| Pre-alb (n = 169) | 5.9 (3.3–10.0) | 3.7 (2.5–6.4) | 4.8 (2.6–8.9) | 0.040 |
| alb (n = 171) | 1.8 (1.4–2.2) | 1.5 (1.2–2.0) | 1.5 (1.2–1.8) | 0.004 |

IQR, interquartile range; AST; Aspartate transaminase, ALT; Alanine transaminase, γ-GTP; γ-glutamyl transpeptidase, ALP; alkaline phosphatase, UIBC; Unsaturated iron binding capacity, Fe; Serum iron, IL-6; Interleukin 6, Pre-alb; pre-albumin, alb; albumin.

large intestine, there is a risk of iron accumulation, even with oral intake [9]. If oral consumption of iron exceeds 100 mg/day, iron overload (Bantu siderosis) can occur [12].

In patients with difficulty consuming iron orally, long-term TPN rarely leads to iron deficiency, and the American Society for Parenteral and Enteral Nutrition (ASPEN) 2002 guidelines [10] do not recommend upper and lower limits for the amount of iron infused per day. However, several studies have reported that in adults and children receiving TPN, 29%–55% [11] and 22% [12], respectively, experienced anaemia. The ASPEN guidelines were therefore revised to recommend an iron dose of 1–5 mg/day [2]. Similarly, the European Society for Clinical Nutrition and Metabolism (ESPEN) guidelines recommend a dose of 1–1.2 mg/day of iron for parental nutrition [13]. In Japan, the recommended dose of iron with TPN is 2 mg

**Table 6. Factors associated with iron staining using logistic regression models.**

| | coefficient | SE | *P* value | OR | 95% CI |
|---|---|---|---|---|---|
| **Model 1** | | | | | |
| Constant | -1.556 | 0.537 | | | |
| Male gender | 1.416 | 0.430 | 0.001 | 4.119 | 1.773–9.566 |
| Iron in post-mortem serum | 0.025 | 0.008 | 0.001 | 1.025 | 1.010–1.041 |
| **Model 2** | | | | | |
| Constant | -2.688 | 0.561 | | | |
| UIBC in post-mortem serum | 0.014 | 0.004 | 0.000 | 1.014 | 1.006–1.022 |
| TPN | 1.277 | 0.392 | 0.001 | 3.586 | 1.664–7.732 |

SE, standard error; OR, odds ratio; CI, confidence interval; UIBC, unsaturated iron binding capacity; TPN, total parenteral nutrition

Adjusted covariance of Model 1: age, gender, history of operation, history of fracture, TPN, partial parenteral nutrition (PPN), number of days of TPN and volume of iron during the year prior to the patients' death, number of days of TPN and amount of iron during the entire period of hospitalization, serum UIBC, serum ferritin, and serum iron.

Covariance of Model 2: age, gender, history of operation, history of fracture, TPN, PPN, number of days of TPN for one year prior to the patients' death and during the entire period of hospitalization, serum UIBC, serum ferritin, and serum iron, serum UIBC, serum ferritin, and serum iron.

(35 μmol)/day, and the iron concentration in blended mineral preparations is 2 mg per ampoule (A). To avoid iron overload, a new formula of TPN using an all-in-one bag has an iron dose of 1.1 mg/bag, regardless of volume or calories. However, these recommendations are based on predicted consumption, transitions in serum iron, and haemoglobin levels. Although the data are no longer current, a report from Japan found that no side effects were caused by the administration of 10 mg ferric chloride (about 2 mg of iron) per day in a 54-year-old male receiving TPN for four years [14].

The criteria for iron overload are based on ferritin levels, according to the World Health Organization in 2013 [15]. However, iron overload is often assessed by haemoglobin and serum iron levels. Even ageing anaemic patients with adequate ferritin levels, but in a state of chronic inflammation, may be inadvertently administered iron [16]. It is unclear whether the amount of iron administered is appropriate, but there are several reports of iron overload using MRI and liver fibrosis markers in cases of thalassemia [17, 18]. Furthermore, in hepatitis C cases, associations of iron overload in liver biopsies have also been investigated [17]. However, these data apply to specific diseases, and do not specify cases of parenteral iron infusion.

Iron deposition in the liver has not been previously studied in autopsied patients, and in our study, intravenous iron administration has been our focus. Because our study used retrospective data from clinical records, the exact daily doses and dosing periods could not be precisely assessed. In contrast, it is impossible to perform prospective studies on human liver tissue. In our case, the maximum iron dose was estimated from the administration of an integrated trace element preparation, 1 A (ferric chloride 9.460 mg/Fe 35 μmol or 2 mg), present in only a few cases. Most patients were given 2–3 A/week, or ferric chloride 4.73 mg/Fe 17.5 μmol (1 mg)/day in a pre-packaged blend. This amount adheres to the lower recommended limit. Based on our assessment of the medical records, the number of days, and total amount of iron administration, there were no cases of patients receiving more than 35 μmol of iron/day. In this study, haemoglobin level of many cases was a little below the normal level. But none of the cases had intensive intravenous iron administration for therapeutic purposes under the diagnosis of iron deficiency anemia. In almost all cases, an all-in-one bag TPN formulation containing a mineral component (iron dose of 1.1 mg/bag) was used for daily iron supplementation.

In this study, over half of the patients were positive for iron deposits in the liver. Prussian blue staining [19] highlights the iron-storage complex haemosiderin, non-haeme iron ($Fe^{3+}$), and severe iron deposition in haemosiderosis [20]. This form of iron, $Fe^{3+}$, is involved in the generation of free radicals by the Fenton reaction, which causes severe cellular and tissue damage, thereby contributing to fibrosis of the liver [21]. In a study of hepatitis C, a strong correlation has been found between iron deposition and fibrosis of the liver [22].

First, it is worth noting that males showed more iron deposition than females (Table 1). This tendency has been described previously in hepatitis C patients in a study that found that liver iron accumulation was associated with the male sex, reflected by high serum iron and ferritin levels [23].

With iron administration and iron deposition in the liver, it is impossible to know the exact lifetime dose of iron. In our paper, we focused on the intake of iron by infusion, although there are also reported cases in which secondary haemochromatosis is caused by oral iron administration [24]. Additionally, regardless of the aetiology, iron-loading is frequently observed in chronic liver diseases, such as iron overload syndrome, hereditary haemochromatosis, viral hepatitis, alcoholic liver disease, non-alcoholic fatty liver disease, non-alcoholic steatohepatitis, and diabetes [21]. Abnormalities in iron metabolism have also been reported in patients with acquired immunodeficiency syndrome (AIDS) [25].

In this study, no patients were previously diagnosed with liver disease, although there were multiple cases of organ and liver failure at the time of death. In assessing the route of nutritional administration, the percentage of patients on TPN (82.1%) was significantly higher in the iron-positive group. Additionally, this group appeared to have a much higher volume of iron administered in the year before death compared with the iron-negative group. The maximum amount of iron administered, 3,535 μmol/year, was nearly equivalent to 10 μmol (0.57 mg)/day and less than the recommended dose. Although it was impossible to obtain the exact dosage, the possibility of iron deposition in the liver cannot be eliminated in long-term TPN patients. We recommend that the amount of intravenous iron administered be carefully evaluated, especially in older adult patients.

Many physicians still prescribe iron supplementation based solely on haemoglobin and serum iron indices. However, ferritin should also be checked in all patients, and not only in those with chronic kidney disease, as previously mentioned in reviews [26]. In our post-mortem blood samples, ferritin levels correlated strongly with liver iron deposition. However, the ferritin level was higher in some patients without iron staining than those with iron staining. In patients with chronic inflammatory diseases, higher ferritin levels are more likely to be caused by problems in iron metabolism whether there is a lot of iron or not. That is, regardless of the accumulation of iron [27]. In the present study, all patients had been hospitalized with terminal diseases, and their median age was 85 years. Among some of these patients, the ferritin level may not be related to iron deposition in the liver. In contrast, UIBC was significantly lower, and serum iron and ferritin were significantly higher in the iron-positive group. No association was found with hepcidin and IL-6, which are key factors in iron metabolism.

Our study showed that long-term iron infusion can lead to iron deposition in the liver. It is possible that iron is deposited in other tissues as well, although we did not investigate this. We found that iron deposition in the liver was strongly correlated with ferritin levels. Therefore, ferritin should be considered a good marker for iron.

The recommended dose of iron in Japan is slightly higher than that in Europe and the United States. In this study, iron deposition in the liver was examined, and chronic intravenous iron administration was associated with tissue deposition of iron, even at less than 10 μmol (0.57 mg)/day. This suggests that tissue iron deposition occurs in long-term TPN patients, even when the daily dose of iron is less than the recommended amount.

The limitations of the present study are associated with its retrospective design. Although we are the first to demonstrate a strong correlation between the long-term use of TPN and iron deposition in the liver, we do not fully understand the cause or mechanism of iron accumulation. Moreover, we were unable to obtain accurate data on the dosage of intravenous iron administration before hospital admission. Nevertheless, our results provide important information for physicians caring for older patients using long-term TPN.

## Conclusions

Chronic intravenous administration of iron was associated with iron deposition in the liver, even in patients receiving the minimum recommended amount. Therefore, in long-term TPN patients, iron dose should be carefully considered. The results of the present study highlight the value of analyzing and evaluating post-mortem samples to potentially improve the future treatment and care of elderly patients.

## Supporting information

**S1 Data file.**
(XLSX)

## Acknowledgments

We thank all our patients and their families for agreeing to an autopsy and participating in our research study. We also thank the staff of Fukushimura Hospital for assisting with the medical records. We gratefully acknowledge medical students Fubuki Inoue, Naoto Imura, and Hideshi Hibi for their help in processing and staining pathological samples.

## Author Contributions

**Conceptualization:** Hiroyasu Akatsu, Toshie Manabe.

**Data curation:** Hiroyasu Akatsu, Yoshihiro Kawade, Hajime Tanaka, Takayoshi Kanematsu, Kazuyuki Arakawa, Yoshiyuki Masaki, Chie Hishida, Takeshi Kanesaka, Norihiro Ogawa, Yoshio Hashizume, Koichi Tsuneyama.

**Formal analysis:** Toshie Manabe.

**Funding acquisition:** Hiroyasu Akatsu.

**Investigation:** Hiroyasu Akatsu, Toshie Manabe, Yoshihiro Kawade, Hajime Tanaka, Takayoshi Kanematsu, Kazuyuki Arakawa, Yoshiyuki Masaki, Chie Hishida, Takeshi Kanesaka, Norihiro Ogawa, Yoshio Hashizume, Koichi Tsuneyama, Hirotaka Ohara, Mitsuo Maruyama, Takayuki Yamamoto.

**Project administration:** Hirotaka Ohara, Takayuki Yamamoto.

**Supervision:** Hiroyasu Akatsu, Hirotaka Ohara, Mitsuo Maruyama.

**Writing – original draft:** Hiroyasu Akatsu, Toshie Manabe.

**Writing – review & editing:** Hiroyasu Akatsu, Toshie Manabe, Yoshihiro Kawade, Hajime Tanaka, Takayoshi Kanematsu, Kazuyuki Arakawa, Yoshiyuki Masaki, Chie Hishida, Takeshi Kanesaka, Norihiro Ogawa, Yoshio Hashizume, Koichi Tsuneyama, Hirotaka Ohara, Mitsuo Maruyama, Takayuki Yamamoto.

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
