## [Decision Letter · Decision Letter 0]

1 Apr 2020

PONE-D-20-05549

Iron deposition in autopsied liver specimens from older patients receiving intravenous iron infusion in terminal

PLOS ONE

Dear Prof. Hiroyasu Akatsu,

Thank you for submitting your manuscript to PLOS ONE. After careful consideration, we feel that it has merit but does not fully meet PLOS ONE’s publication criteria as it currently stands. Therefore, we invite you to submit a revised version of the manuscript that addresses the points raised during the review process.

We would appreciate receiving your revised manuscript by May 16 2020 11:59PM. To enhance the reproducibility of your results, we recommend that if applicable you deposit your laboratory protocols in protocols.io, where a protocol can be assigned its own identifier (DOI) such that it can be cited independently in the future. For instructions see: http://journals.plos.org/plosone/s/submission-guidelines#loc-laboratory-protocols

We look forward to receiving your revised manuscript.

Kind regards,

Tatsuo Kanda, M.D., Ph.D.

Academic Editor

PLOS ONE

Journal Requirements:

2. As mentioned in your Limitations section, causation  cannot be determined by retrospective studies; thus, please revise any causality statement made in the Abstract and Discussion sections.

3. Please confirm if the title is correct, should it be " Iron deposition in autopsied liver specimens from older patients receiving intravenous iron infusion"

4. Thank you for stating the following financial disclosure: "The authors received no specific funding for this work."

Reviewers' comments:

Reviewer's Responses to Questions

**Comments to the Author**

1. Is the manuscript technically sound, and do the data support the conclusions?

Reviewer #1: Yes

Reviewer #2: Yes

2. Has the statistical analysis been performed appropriately and rigorously? 

Reviewer #1: Yes

Reviewer #2: No

3. Have the authors made all data underlying the findings in their manuscript fully available?

Reviewer #1: Yes

Reviewer #2: Yes

4. Is the manuscript presented in an intelligible fashion and written in standard English?

Reviewer #1: No

Reviewer #2: Yes

5. Review Comments to the Author

Reviewer #1: In the present study, authors investigated iron deposition in autopsied live specimens from patients receiving intravenous iron infusion. They concluded that iron administration by long-term total parenteral nutrition (TPN) affected the iron deposition in liver. Although this is a well-written paper that presents interesting data, there are several drawbacks in the manuscript

Major points

1) To the best of my knowledge, this is the first study to demonstrate the association between iron administration and iron deposition in liver specimens. I think the authors should place more emphasis on the point in both the background and discussion section.

2) Regrettably, the authors simply indicated conditions of iron administration in terms of with or without iron staining (Table 3), I think it would be useful if the authors gave more data according to the iron staining grade.

3) The authors state that clinical data on iron administration including volume, data, periods and frequency were assessed in the method section. Analysis of the frequency of iron administration should be added to the result section.

4) Page 7 Line 103: ‘At least (three independent investigators)’ is unclear. Can the number of investigators differ according to each cases? Moreover, was there an intraobserver error? These points should be explained.

5) Bowel resection could be related to the iron overload. In the table 1, iron-positive patients were more likely to have a history of surgery. The authors should explain in more detail the data and further discussion needs to be included to address the point.

6) What is the best predictable marker for iron deposition in terms of the iron staining in autopsied liver specimens? I think the quality of this paper could be enhanced by adding the data.

7) There are quite some odd wording and use of grammar. English of the manuscript has to be checked by a native speaker.

Minor points

1) In the abstract section, the authors should definitely describe the purpose of the study.

2) In the abstract section, storage condition of the sample was unnecessary.

3) Page11, Line167; ’81.2%’ should be changed into ’82.1%’.

Reviewer #2: 1. Line 275, “3.54 μmol/year or 10 μmol (1 mg)/day”

Why iron amount per year is lower than that per day?

2. Line 246, “Fe 17.5 μmol (1 mg)/day”

Line 275, “10 μmol (1 mg)/day”

Line 296, “10 mmol (1 mg)/day.” Which is correct?

3. Table 3, “Iron given 1 year before death (μmol L).” “Total iron volume during inpatient infusion (μmol L) “

Please explain the unit “μmol L”. I think total iron amount administerd can be described as mass rather than concentration.

4. In table 4 and 5, why did total cohort including patients without iron staining in the liver show high ferritin level?

5. Multivariate analysis can be performed using factors in Table 3 and 4.

6. Please add the factors of TPN and iron infusion in Table 5.

6. PLOS authors have the option to publish the peer review history of their article (what does this mean?). If published, this will include your full peer review and any attached files.

Reviewer #1: No

Reviewer #2: No

---

## [Author Response · Author response to Decision Letter 0]

25 May 2020

Point-by-point response to reviewers’ comments

Thank you very much for your time and valuable feedback. We have carefully considered your comments and suggestions and have revised our manuscript accordingly. We have included our responses to the reviewers’ comments below. For clarity, the reviewer’s comments are shown in blue and our responses are shown in black.

Reviewer reports:

Reviewer #1:

In the present study, authors investigated iron deposition in autopsied live specimens from patients receiving intravenous iron infusion. They concluded that iron administration by long-term total parenteral nutrition (TPN) affected the iron deposition in liver. Although this is a well-written paper that presents interesting data, there are several drawbacks in the manuscript

Major points

1) To the best of my knowledge, this is the first study to demonstrate the association between iron administration and iron deposition in liver specimens. I think the authors should place more emphasis on the point in both the background and discussion section.

Thank you very much for your kind comment and pointing out the significance of our work. Yes, we also believe that this is the first study demonstrating the association between iron administration and iron deposition in the liver, and we have added statements to the Introduction and Discussion sections emphasising this point. 

P. 5; Lines 61-62.

However, the association between iron administration by TPN and iron deposition in the liver has never been examined. 

P. 15; Lines 241-242. 

This is the first study to demonstrate an association between iron administration and iron deposition in post-mortem liver specimens.

2) Regrettably, the authors simply indicated conditions of iron administration in terms of with or without iron staining (Table 3), I think it would be useful if the authors gave more data according to the iron staining grade.

Thank you very much for this important suggestion. As you mentioned in comment #4, however, the grading of iron staining was variable, even among our expert observers. For this reason, we used positive or negative iron staining as the principal comparison in Tables 1–4. However, in Table 5, we have presented our laboratory findings in post-mortem serum according to the grade of iron staining. We hope this is sufficient.

3) The authors state that clinical data on iron administration including volume, data, periods and frequency were assessed in the method section. Analysis of the frequency of iron administration should be added to the result section.

As suggested, we have added the method of data collection for iron administration to the methods section and have modified our explanation of the data in the Results section. 

P. 5; Lines 75-77.

Data on the number of days and the volume of iron administration were collected for one year prior to the patients’ death and during the entire period of hospitalization. 

P. 12; Lines 188-192.

The number of days of TPN administration was significantly higher in the iron-positive group compared with the iron-negative group both in the year before death (p = 0.004) and during hospitalization (p = 0.038). However, the volume of TPN was not significantly different between the groups in the year before death (p = 0.222) or during the hospitalization (p = 0.220). 

4) Page 7 Line 103: ‘At least (three independent investigators)’ is unclear. Can the number of investigators differ according to each cases? Moreover, was there an intraobserver error? These points should be explained.

The “three independent investigators” are co-authors of the manuscript. We have identified the investigators using their initials and have added an explanation of how we minimised interobserver error.

P. 7; Line 104-108.

Three investigators (H.A., K.T., H.O.), who specialize in the study of liver or liver pathology, evaluated the Prussian blue staining in each liver sample independently and recorded their findings on a custom data collection form. We verified the accuracy of these evaluations by comparing the forms of each investigator and resolving any discrepancies through discussion. 

5) Bowel resection could be related to the iron overload. In the table 1, iron-positive patients were more likely to have a history of surgery. The authors should explain in more detail the data and further discussion needs to be included to address the point.

P. 9; Lines 144-149.

The most frequent cases of surgery were for femoral fracture (10 cases), female reproductive ailments (7 cases) or digestive system disorders outside of the intestine (7 cases). There were two cases of bowel resection for colorectal cancer, and both patients had an iron deposition grade of I or II in the liver. Patients with a history of fractures, other than a femoral fracture, were more likely to be iron-negative.

6) What is the best predictable marker for iron deposition in terms of the iron staining in autopsied liver specimens? I think the quality of this paper could be enhanced by adding the data.

In the Results section, we have added an analysis of predictable markers for iron deposition using logistic regression models. These results are presented in Table 6 with the subheading, “Factors associated with iron staining.”

P. 12; Lines 218-224.

Logistic regression analysis revealed that male sex (odds ratio [OR], 4.119; 95% confidence interval [CI], 1.773-9.566) and iron in post-mortem serum (OR, 1.025; 95% CI, 1.010-1.041) were factors associated with iron staining if the volume of TPN was included in the adjusted variables. If the volume of TPN was excluded from this analysis, UIBC in post-mortem serum (OR, 1.014; 95% CI, 1.003-1.022) and administration of TPN (OR, 3.586; 95% CI, 1.664-7.732) were factors significantly associated with iron staining.

7) There are quite some odd wording and use of grammar. English of the manuscript has to be checked by a native speaker.

Our manuscript was revised by professional scientific editors who are native English speakers. 

Minor points

1) In the abstract section, the authors should definitely describe the purpose of the study.

We have added the purpose of the study to the abstract.

P. 2; Lines 6-8.

The aim of the study was to assess the influence of iron administration by long-term TPN on iron deposition in post-mortem liver samples isolated from older deceased patients.

2) In the abstract section, storage condition of the sample was unnecessary.

We have revised this sentence.

P. 2; Lines 15-17.

Samples were collected for the measurement of unsaturated serum iron, serum iron, albumin, prealbumin, hepcidin, and IL-6 concentrations. 

3) Page11, Line167; ’81.2%’ should be changed into ’82.1%’.

We have corrected 81.2% to 82.1%.

P. 11; Line 184.

……in the iron-positive group (82.1%; p<0.001)

Reviewer #2: 

1. Line 275, “3.54 μmol/year or 10 μmol (1 mg)/day”

Why iron amount per year is lower than that per day?

Thank you for pointing this out. It was a mistake, and we have corrected it in the text.

P. 19; Lines 316-317.

The maximum amount of iron administered, 3,535 μmol/year, was nearly equivalent to 10 μmol (0.57 mg)/day

2. Line 246, “Fe 17.5 μmol (1 mg)/day”

Line 275, “10 μmol (1 mg)/day”

Line 296, “10 mmol (1 mg)/day.” Which is correct?

“Fe 17.5 μmol (1 mg)/day” is correct. The others have been changed.

3. Table 3, “Iron given 1 year before death (μmol L).” “Total iron volume during inpatient infusion (μmol L) “

Please explain the unit “μmol L”. I think total iron amount administerd can be described as mass rather than concentration.

Thank you for your comment. This was also an error. The “L” has been deleted.

4. In table 4 and 5, why did total cohort including patients without iron staining in the liver show high ferritin level?

In patients with chronic inflammatory disease, the higher ferritin levels are likely to be caused by problems in iron metabolism rather than a reflection of iron deposition from TPN. In the present study, all patients were hospitalised and at the end stages of their lives. We believe that the ferritin levels in these particular patients are not related to iron deposition in the liver. We have therefore included an explanation for this in the Discussion section and have added reference 27.

P. 20; Lines 325-331.

However, the ferritin level was higher in some patients without iron staining than those with iron staining. In patients with chronic inflammatory diseases, higher ferritin levels are more likely to be caused by problems in iron metabolism whether there is a lot of iron or not, that is, regardless of the accumulation of iron [27]. In the present study, all patients had been hospitalized with terminal diseases, and their median age was 85 years. Among some of these patients, the ferritin level may not be related to iron deposition in the liver. 

5. Multivariate analysis can be performed using factors in Table 3 and 4.

6. Please add the factors of TPN and iron infusion in Table 5.

Based on your suggestions in comments #5 and #6, we have analysed factors associated with iron staining using logistic regression models. These results are presented in Table 6 of the Results section with the subheading “Factors associated with iron staining.”

P. 14; Lines 218-224.

Logistic regression analysis revealed that male sex (odds ratio [OR], 4.119; 95% confidence interval [CI], 1.773-9.566) and iron in post-mortem serum (OR, 1.025; 95% CI, 1.010-1.041) were factors associated with iron staining if the volume of TPN was included in the adjusted variables. If the volume of TPN was excluded from this analysis, UIBC in post-mortem serum (OR, 1.014; 95% CI, 1.003-1.022) and administration of TPN (OR, 3.586; 95% CI, 1.664-7.732) were factors significantly associated with iron staining.

---

## [Decision Letter · Decision Letter 1]

8 Jun 2020

PONE-D-20-05549R1

Iron deposition in autopsied liver specimens from older patients receiving intravenous iron infusion

PLOS ONE

Dear Dr. Hiroyasu Akatsu,

Thank you for submitting your manuscript to PLOS ONE. After careful consideration, we feel that it has merit but does not fully meet PLOS ONE’s publication criteria as it currently stands. Therefore, we invite you to submit a revised version of the manuscript that addresses the points raised during the review process.

ACADEMIC EDITOR:  Please give some comments to iron drugs (Reviewer 3)

We look forward to receiving your revised manuscript.

Kind regards,

Tatsuo Kanda, M.D., Ph.D.

Academic Editor

PLOS ONE

Reviewers' comments:

Reviewer's Responses to Questions

**Comments to the Author**

1. If the authors have adequately addressed your comments raised in a previous round of review and you feel that this manuscript is now acceptable for publication, you may indicate that here to bypass the “Comments to the Author” section, enter your conflict of interest statement in the “Confidential to Editor” section, and submit your "Accept" recommendation.

Reviewer #1: All comments have been addressed

Reviewer #2: (No Response)

Reviewer #3: (No Response)

2. Is the manuscript technically sound, and do the data support the conclusions?

Reviewer #1: Yes

Reviewer #2: (No Response)

Reviewer #3: Partly

3. Has the statistical analysis been performed appropriately and rigorously? 

Reviewer #1: Yes

Reviewer #2: (No Response)

Reviewer #3: I Don't Know

4. Have the authors made all data underlying the findings in their manuscript fully available?

Reviewer #1: Yes

Reviewer #2: (No Response)

Reviewer #3: Yes

5. Is the manuscript presented in an intelligible fashion and written in standard English?

Reviewer #1: Yes

Reviewer #2: (No Response)

Reviewer #3: Yes

6. Review Comments to the Author

Reviewer #1: Thank you for revising the problem.

The manuscript has been much improved according to the review comments. Furthermore, the second version had been enhanced by adding an analysis of predictable markers for iron deposition using multivariate analysis.

Reviewer #2: (No Response)

Reviewer #3: It is a very interesting paper that warns of iron overdose due to the iron content in high calorie infusion formulations. It is unknown how many people were receiving iron drugs for anemia. Give some comments about iron drugs in discussion section.

7. PLOS authors have the option to publish the peer review history of their article (what does this mean?). If published, this will include your full peer review and any attached files.

Reviewer #1: No

Reviewer #2: No

Reviewer #3: Yes: Hidehiro Kamezaki

---

## [Author Response · Author response to Decision Letter 1]

15 Jul 2020

Point-by-point response to reviewers’ comments

Thank you very much for your time and valuable feedback. We have carefully considered your comments and suggestions and have revised our manuscript accordingly. We have included our responses to the reviewers’ comments below. For clarity, the reviewer’s comments are shown in blue and our responses are shown in black.

Reviewer reports:

Reviewer #1: Thank you for revising the problem.

The manuscript has been much improved according to the review comments. Furthermore, the second version had been enhanced by adding an analysis of predictable markers for iron deposition using multivariate analysis.

Thank you very much.

Reviewer #3: It is a very interesting paper that warns of iron overdose due to the iron content in high calorie infusion formulations. It is unknown how many people were receiving iron drugs for anemia. Give some comments about iron drugs in discussion section.

Thank you very much for your comments. According to the medical record, none of the patients were receiving iron drug for the anemia. Basically, the hemoglobin level of the elderly is often lower than the normal value, but this is because of iron utilization disorder in the terminal elderly. The attending physician did not adjust the iron preparation due to some fluctuation in hemoglobin. In addition, they had not checked strictly the serum iron, unsaturated iron binding capacity, and ferritin level. The iron agent contained in daily TPN is the purpose of daily iron supplementation rather than the purpose of improvement. In addition, there were no cases where iron was administered to improve the anemia. We added and described these points.

P. 18; Lines 290-294.

In this study, haemoglobin level of many cases was a little below the normal level. But none of the cases had intensive intravenous iron administration for therapeutic purposes under the diagnosis of iron deficiency anemia. In almost all cases, an all-in-one bag TPN formulation containing a mineral component (iron dose of 1.1 mg/bag) was used for daily iron supplementation.

After checked in-house

1. Please ensure that you refer to Table 6 in your text as, if accepted, production will need this reference to link the reader to the Table.

P. 14; Lines 221, and 224

Thank you very much for the advice. I have referred to Table 6 in text at P.14; Lines 221, and 224.

---

## [Decision Letter · Decision Letter 2]

21 Jul 2020

Iron deposition in autopsied liver specimens from older patients receiving intravenous iron infusion

PONE-D-20-05549R2

Dear Dr. Hiroyasu Akatsu,

We’re pleased to inform you that your manuscript has been judged scientifically suitable for publication and will be formally accepted for publication once it meets all outstanding technical requirements.

Kind regards,

Tatsuo Kanda, M.D., Ph.D.

Academic Editor

PLOS ONE

Additional Editor Comments (optional):

Reviewers' comments:

Reviewer's Responses to Questions

**Comments to the Author**

1. If the authors have adequately addressed your comments raised in a previous round of review and you feel that this manuscript is now acceptable for publication, you may indicate that here to bypass the “Comments to the Author” section, enter your conflict of interest statement in the “Confidential to Editor” section, and submit your "Accept" recommendation.

Reviewer #2: (No Response)

Reviewer #3: All comments have been addressed

2. Is the manuscript technically sound, and do the data support the conclusions?

Reviewer #2: (No Response)

Reviewer #3: Yes

3. Has the statistical analysis been performed appropriately and rigorously? 

Reviewer #2: (No Response)

Reviewer #3: Yes

4. Have the authors made all data underlying the findings in their manuscript fully available?

Reviewer #2: (No Response)

Reviewer #3: Yes

5. Is the manuscript presented in an intelligible fashion and written in standard English?

Reviewer #2: (No Response)

Reviewer #3: Yes

6. Review Comments to the Author

Reviewer #2: (No Response)

Reviewer #3: (No Response)

7. PLOS authors have the option to publish the peer review history of their article (what does this mean?). If published, this will include your full peer review and any attached files.

Reviewer #2: No

Reviewer #3: **Yes: **Hidehiro Kamezaki

---

## [Editor Report · Acceptance letter]

23 Jul 2020

PONE-D-20-05549R2 

 Iron deposition in autopsied liver specimens from older patients receiving intravenous iron infusion 

Dear Dr. Akatsu:

I'm pleased to inform you that your manuscript has been deemed suitable for publication in PLOS ONE. Congratulations! Your manuscript is now with our production department. 

Kind regards, 

on behalf of

Dr. Tatsuo Kanda 

Academic Editor

PLOS ONE